# Formability, Microstructure and Properties of Inconel 718 Superalloy Fabricated by Selective Laser Melting Additive Manufacture Technology

**DOI:** 10.3390/ma14040991

**Published:** 2021-02-19

**Authors:** Xiaoping Liu, Kuaishe Wang, Ping Hu, Xiaomei He, Baicheng Yan, Xuzhao Zhao

**Affiliations:** 1School of Metallurgical Engineering, Xi’an University of Architecture and Technology, Xi’an 710055, China; huping1985@126.com (P.H.); xmhe1980@126.com (X.H.); bcyan1998@163.com (B.Y.); xuzzhao@163.com (X.Z.); 2National and Local Joint Engineering Research Center for Functional Materials Processing, Xi’an University of Architecture and Technology, Xi’an 710055, China; 3Key Laboratory of Gold and Resource of Shaanxi Province, School of Metallurgical Engineering, Xi’an University of Architecture and Technology, Xi’an 710055, China

**Keywords:** Inconel 718, selective laser melting (SLM), densification, microstructure, mechanical property

## Abstract

Many urgently needed inconel superalloy parts with complex internal cavity geometry and high surface precision are difficult to prepare by traditional subtractive manufacturing methods because of its poor machinability. The additive manufacturing technology that has emerged in recent years became a research hotspot in the manufacture of refractory and difficult-to-process metals. In the present study, selective laser melting (SLM), a typical additive manufacture technology, was used to prepare Inconel 718 samples. The influences of input laser energy density ((E, J/mm^3^) on densification behavior, phases composition, microstructures, microhardness, and wear performance of the SLM as-built Inconel 718 samples were explored in detail. X-ray diffraction (XRD), optical microscopy (OM), scanning electron microscopy (SEM), and transmission electron microscopy (TEM) were used to examine the phase composition and microstructure evolutions. The results show that the formablity, microstructures and mechanical properties of the printed samples were all improved with the increase of E within the parameter setting range of this study. At a lower E, the poor surface morphology and balling effect occurred, the density, hardness, and wear resistance were all at a relatively lower level. When an E value of 190 J/mm was properly set, the surface open-pores and balling effect disappeared, the laser scanning tracks became smooth and continuous, the near-full dense (99.15%) and specimens with good metallurgical bonding and no critical defect were obtained, in which the average microhardness value reached 348 HV_0.2_ and wear rate was 5.67 × 10^−4^ mm^3^/N·m. The homogeneity of the superalloy Inconel 718 was also explored.

## 1. Introduction

In the past four decades, nickel-base superalloys have experienced rapid development and received widespread attention from experts in various fields [1,2,3]. Among them, Inconel 718, a typical austenitic superalloy with Ni, Cr, Fe, Mo, Co, Nb as the main elements, has been widely used in the fields such as aeronautics and astronautics, guide vanes, and high-pressure compressor discs in gas turbines due to its excellent comprehensive mechanical properties, sound corrosion resistance and high oxidation resistance under high temperature conditions [4,5,6]. With the development of modern industry, new advanced equipment has put forward higher and higher requirements on the shape, structure and performance of components, and the demand for Inconel 718 parts with complex geometries, especially complex internal cavity geometry and porous structure with high dimension precision is becoming more and more urgent. However, its poor machinability and difficulty to control of casting and forging processes make it hard to adapt the traditional subtractive manufacturing because it tends to cause severe tool over-wear and high surface roughness which affects the dimensional accuracy of the part [7,8,9,10,11,12]. As one of the most promising additive manufacturing technology, selective laser melting (SLM) technology is expected to solve this problem due to its unique ability of rapid production of three-dimensional (3D) parts without geometric restrictions and the material waste [13,14,15,16].

During SLM, 3D geometrically dense complex components can be fabricated directly by selectively melting the pre-spread powder layer by layer under the irradiation of high-energy laser [17,18,19]. With the characteristics such as thin powder layer, small laser spot, and fast scanning speed, SLM can fabricate near-full dense samples with geometrically complex structure and sound surface integrity without post-treatment, which cannot be easily achieved by traditional methods such as forging and casting [20]. Furthermore, the unique dendritic microstructures of the SLM as-built samples can be dominated by the complex physical and chemical behaviors within the laser micro-melting pool because the SLM technology based on the nonequilibrium extremely rapid melting/solidification mechanism. Its high localized thermal gradients and rapid solidification processes provide the potential for obtaining non-equilibrium fine-grained microstructures and novel metallurgical properties, making SLM research a hot topic [21].

Many experimental studies have focused on the microstructures and properties of the SLM-formed Inconel 718. Tao et al. [22] focused on the crystal growth process during SLM Inconel 718 and pointed out that the grains grew in a cellar-dendritic pattern. StrdRner et al. [23] found the microstructures of SLM-fabricated Inconel 718 was very fine and oriented along the building direction. Yi et al. [24] found there was a bright crescent-shaped region at the bottom of the SLM-Inconel 718 melt pool consisting of finer columnar dendrites, which had the same orientation as the previous melt pool. As for the effects of building parameters, Deng et al. [25] pointed out that when the input laser energy exceeded the limit, “balling phenomenon” will occur at the molten pool. Wan et al. [26] investigated the influence of scanning strategy on microstructure and texture of SLM-fabricated Inconel 718. Popovich et al. [27] proved the ability of additive manufacturing to produce customized microstructures and mechanical properties. Apart from these studies, many scholars focused on kinds of post heat-treatments such as solution treatment, annealing and hot isostatic pressing (HIP)) to improve the comprehensive performance of SLM-Inconel 718 parts. Zhao et al. [28] found that the tensile strength of the heat-treated laser rapid forming Inconel 718 was comparable with the wrought Inconel 718 alloy, which was 1.5 times that of the as-built samples. Amato et al. [10] pointed out that the microstructure of the SLM-formed part exhibited columnar grains whether it was parallel to or perpendicular to the building direction, microindentation (Vickers) hardness of as-fabricated, HIP and annealed material were investigated.

Few studies on the influence of laser parameters on the SLM-parts’ microstructure and mechanical properties have been reported [29]. Among the previous studies of mechanical properties, there is a gap in the research on the wear resistance of SLM-Inconel 718, which urgently needs to be filled due to the severe working conditions. As a novel net-shape manufacturing method, the research on Inconel 718 SLM processing at home and abroad is not perfect due to its extremely rapid and complicated processing conditions. In order to meet the practical application of SLM-formed parts in the industrial field, it is still necessary to conduct in-depth discussions on the optimization of process parameters, forming process control and metallurgical mechanisms. In the present study, Inconel 718 specimens were prepared by SLM under different laser processing parameters. The influence of input laser energy density (E) on surface morphologies, densities, phases, microstructures, and mechanical behaviors were investigated. The relationship between input laser energy density, density, microstructures and mechanical performances of Inconel 718 parts was established. The internal physical and chemical mechanisms for the evolution of microstructural and mechanical performance during SLM can also provide the theoretical basis for selective laser melting of other materials or other laser powder processing methods such as laser engineered net shaping and laser metal deposition.

## 2. Materials and Methods

### 2.1. Raw Materials

In this work, the spherical gas-atomized Inconel 718 superalloy powders were used as the raw material, whose particle size ranged from 10–50 μm. The chemical compositions of the main elements of Inconel 718 powder are shown in Table 1. Figure 1 is the SEM image of Inconel 718 raw material powder.

### 2.2. SLM Processing

The SLM experiment was performed on the a Mlab Cusing R 3D printing system (Concept Laser, Upper Franconia, Lichtenfels, Germany) which mainly consists of a 100 W continuous wave fiber laser, a computer controlling device, an automatic powder delivery device, and an inert-gas protection device. The SLM forming process was protected by high purity N_2_ gas, maintaining the oxygen level below 100 ppm. Inconel 718 cuboid specimens (9 × 9 × 8 mm^3^) were successfully fabricated by SLM. Figure 2 shows the laser selective melting scanning strategy schematic.

The parameter “laser energy density” E (in J/mm^3^) was defined as the input laser energy to the pre-spread powder per unit volume, as expressed in Equation (1):(1)$$E=\frac{P}{\mathrm{vht}}$$
where *P* the input laser power (W), v the laser scanning speed (mm/s), h the hatch distance (mm), and t the powder layer thickness. The laser energy density E was considered to be a key factor affecting the quality of SLM-formed parts.

According to a series of previous experiments, the setting values of each experimental parameter for SLM-fabrication of Inconel 718 are shown in Table 2. Thus, four different laser energy density of 103, 143, 171, 190 J/mm^3^ were determined to study the influences of laser parameters on formability, microstructure and mechanical properties of the SLM-formed Inconel 718 parts.

### 2.3. Microstructural and Mechanical Properties Characterization

The relative density of the SLM as-built Inconel 718 specimens was examined using an MH-600A solid densitometer (Xiamen Xiongfa Instrument Co., Ltd., Xiamen, China) based on the Archimedes principle. The phase was identified by a D8 Advance A25 X-ray diffractometer (XRD, Bruker, Rheinstetten, Germany) with Cu Kα radiation (λ = 0.1540598 nm) at 40 kV and 40 mA, in a continuous scanning pattern. A fast scan at 4°/min was carried out over 2θ = 20°–100° firstly to give a general exhibition of XRD diffraction peaks. A slower scan rate of 1°/min over 2θ = 42°–45° was further used to obtain more accurate phase identifications. The prepared sample was cut, mounted, pre-ground and polished according to the metallographic standard procedures for microstructural analysis. Deionized water, H_2_O_2_ and HCl were mixed in 2:1:2 volume ratio as a corrosive agent, and the etching time was 25 s. The formability, surface morphology and microstructure of SLM-formed samples were examined by optical microscopy (OM, GX51, Olympus, Tokyo, Japan), and field emission scanning electron microscope (FESEM, SEM 300 Gemini, Oberkauchen, Germany) equipped with energy-dispersive X-ray spectroscopy (EDS). A 0.2 mm thick disc was cut from the SLM-fabricated sample on the X-Y plane by EDM for TEM test. Using the mixture of methanol and perchloric acid in a 9:1 volume ratio as an etchant, the thin foil was prepared by jet electropolishing at a temperature of 90 °C and voltage of 30 V, and then examined by a S-5500 high-resolution field emission SEM (Hitachi, Tokyo, Japan) for scanning transmission electron microscopy (STEM). EDS energy dispersive X-ray spectroscope was used to determine the chemical compositions, in which three times were tested under the same condition, and the average value was the final value of the composition under this condition, and the error was counted.

The measurement of microhardness was performed on the polished cross-section using a 401-MVD microhardness tester with a load of 200 g and a holding time of 15 s to evaluate the microhardness changes with different E. Each sample was tested ten times under the same conditions, and the average was taken as the final hardness value. Dry sliding tests at room temperature were carried out on MS-T3001 ball-disk friction tester to evaluate the wear properties. The counterface material was a stainless steel ball with a diameter of 3 mm with a test load of 500 g. Each friction unit worked for 10 min at a rotation speed of 200 rpm and a radius of 1.5 mm, recording the coefficient of friction (COF) during the dry slide test. The wear volumes (*V*, mm^3^) of specimens were calculated by gravimetric analysis according to the following formula:(2)V=M/ρ
where M (g) was the mass loss of the sample during wear experiments and ρ (g/mm^3^) the density of as-built samples.The wear rate (ω, mm^3^/N·m) were followed by:(3)ω=V/(FL)
where *F* was the contact load (N) and *L* was the sliding distance (m).

## 3. Results and Discussion

### 3.1. Surface Morphologies and Densifications

SEM micrographs for typical upper surface morphology of the Inconel 718 specimens fabricated by SLM under various parameters are shown in Figure 3. Surface morphology integrity varied obviously with the different laser energy densities (E). Figure 4 illustrates the changes in the unetched cross-section optical morphologies and the measured relative densities of corresponding specimens. At a lower E of 103 J/mm^3^, shown in Figure 3a, there were a lot of large-sized metal balls on the upper surface of the specimen. In addition, there is no laser scanning track on the surface, and the density of the formed specimen was only 95.75%, which indicated that for Inconel 718 alloy, the input laser energy density E of 103 J/mm^3^ was too low, resulting in the balling phenomenon. When the input E was increased to 143 J/mm^3^ (Figure 3b), the balling phenomenon was obviously alleviated, but a small amount of dispersed open pores formed by solidification shrinkage appeared on the surface, accompanied by discontinuous laser scanning tracks. The density level was increased to 98.24% as a result. At an even higher E of 171 J/mm^3^, as shown in Figure 3c, it was found that the size and number of metal balls distributed on the surface were significantly reduced, the open-holes were almost disappeared, the laser scanning tracks were clear and the density of the shaped sample increased to 98.67%. When E reached 190 J/mm^3^ there is no any metal balls or open-pores on the upper surface and the laser scanning tracks were clear and tidy, achieving a near-full 99.15% density. The analysis showed that for the SLM forming of Inconel 718 superalloy, with the increasing of input laser energy density E, the surface morphology became smooth, the scanning tracks tended to be clearly visible and tidy, and the density of the sample gradually increased within the scope of the present study.

In the present work, the coordination between the input laser power and the scanning speed is a key prerequisite for ensuring the continuous scanning tracks and smooth dense upper surface morphologies of the formed sample. As the high-energy laser focused on the pre-spread metal powders, a micro-sized molten pool was formed. At higher scanning speed and lower E, the duration of the laser beam on the surface of the molten pool was shortened, so the liquidus temperature in the molten pool was relatively low. The dynamic viscosity μ of a fully liquid SLM is dependent on temperature. The relationship between the liquidus viscosity *μ* and the temperature *T* in the molten pool can be expressed as the following formula [20]:(4)$$\mu =\frac{16}{15}\sqrt{\frac{m}{k_{B}T}\gamma }$$
where γ was the surface tension in the pool liquid, m was the atomic mass, k_B_ was the Boltzmann constant, and *T* was the molten pool temperature. According to the formula, the lower the molten pool temperature *T*, the higher the dynamic viscosity *μ* was. Under relatively high scanning speed, the smooth outward diffusion of the liquid phase was hindered due to the high viscosity of the molten pool liquid phase, leading to the discontinuity of the scanning trajectory and the formation of surface open holes in the present layer. In addition, this also caused a bumpy surface, impeding the swiping of the next layer of powder, which may lead to the bridging of powder particles. Such pores sometimes extended to several layers, reducing the density of the SLM-formed samples. At the same time, the balling effect also promoted the reduction of surface integrity and density level. As the previous literature [30,31,32], the pre-spread metal powder will be completely melted in the present laser parameters, the scanning track can be regarded as a liquid phase cylinder. When satisfying the condition of λ<πD (where *λ* the laser wavelength and *D* the original diameter of the undisturbed liquid cylinder), the stability of the liquid cylinder can be maintained. At a constant *λ*, the combined effect of higher scanning speed and lower laser power input will rapidly reduce the diameter *D* of the liquid phase cylinder, making it impossible to meet the above premise. At this time, the liquid cylinder became more unstable, and the surface energy was reduced by changing the shape into an equilibrium spherical state, resulting in the discontinuity of the scanning tracks on the upper surface of the formed sample. With the increment of the input E, the weakening of the liquid viscosity and the balling effect promoted the densification level to be significantly improved. Therefore, when E was set properly, samples with good surface integrity and near-full density can be obtained.

### 3.2. Phases Compositions

Figure 5a is the general exhibition of XRD diffraction peaks for the SLM-Inconel 718 samples under different E. It can be found that the strong diffraction peaks corresponding to matrix phase γ-Cr-Fe-Ni and precipitating phase γ″-Ni_3_Nb were clearly detected. However, the peak positions of the two phases are overlapping to a large extent. In order to distinguish the phases composition more accurately, further XRD slow-scanning over a narrower range of 2θ = 42°–45° (Figure 5b) was performed. As the input E increased, the location of the diffraction peaks shifted significantly to larger 2θ, and the diffraction peaks of γ and γ″ were obviously broadened and the peak intensity decreased apparently, indicating the smaller lattice parameters of γ-Cr-Fe-Ni with increasing E. The specific value of 2θ location and the peak intensity of γ and γ″ phases are listed in Table 3.

According to Bragg’s law:(5)2dsinθ=nλ(n=1,2,3…)
where d is the lattice planes distance, θ is the diffraction angle, *n* is a constant, and *λ* is the wavelength. As the input E increased, the apparent positive shift of the diffraction peak 2θ indicated a reduction of the lattice planes distance (d), considered to be caused by lattice distortion. It can be inferred that solid solution occurred during the SLM forming process. It is well known that the atomic radii of Ni (1.24 Å), Fe (1.24 Å) and Cr (1.30 Å) are smaller than those of Mo (1.39 Å) and Nb (1.46 Å), while Nb and Mo are the main constituent elements of the Laves phase. When E was increased from 103 J/mm^3^ to 190 J/mm^3^, the temperature of the molten pool was increased and the time that molten pool maintains the liquid state became longer, which was beneficial to the microsegregation of refractory elements such as Nb and Mo, and can promote the formation of the Laves phase in the interdendritic boundaries, as shown in Figure 6a. As a result, the content of Nb and Mo in γ matrix decreased, atoms with smaller atomic radii such as Ni and Fe in the γ matrix occupied the positions of the original large atoms such as Nb and Mo. Therefore, the lattice planes distances of γ matrix became smaller and the diffraction peaks of γ-FeCrNi shifted to a larger 2θ.

To confirm the chemical composition of the irregular material precipitated in the sample, the microstructure on the X-Y plane was further observed and characterized by TEM. The cell structure can be clearly seen from Figure 6a, and each cell structure was pinned with irregular blocky precipitates, and there was a lot of dislocation entanglement around the precipitated phase. Some dislocations can also be clearly seen inside the crystal grains. This was ascribed to the residual stress caused by the rapid cooling process, resulting in the internal structure slipping and deforming to form dislocations. Figure 6b is the selected area electron diffraction (SAED) pattern of γ matrix and massive precipitates. After measurement and calibration, the austenite *γ* matrix has a face-centred cubic (Al) crystal structure, and the constant lattice a = 3.60 Å. The bulk precipitates are Laves phases with the close-packed hexagonal crystal structure, with lattice constants a = 4.81 Å and c = 7.85 Å. Combined with TEM (Figure 6) and STEM (Figure 9) measurements in the X-Y plane, the interdendritic precipitates are identified as Laves phase. The reason that Laves phase hasn’t been detected by XRD may be the limitation of XRD to trace component phase with content less than 5%. 

Rietveld analyses of above the XRD data were carried out using FULLPROF software (2014) to retrieve the structural information of these samples. The Rietveld refinement results are listed in Table 4. It can be seen that the lattice constants of *γ*-matrix phase and *γ*″—precipitate phase all decrease with the increase of E, accompanied by the reduction of *γ*″ weight percentage. With the increase of E, the tendency of Nb and Mo to segregate to the grain boundary becomes stronger, and the content of *γ*″ forming elements Nb and Mo in the matrix becomes smaller, resulting in a decrease in the percentage of *γ*″ and a smaller lattice constant of *γ*-matrix phase and *γ*″—precipitate phase, which is consistent with the above discussion.

### 3.3. Microstructural Characteristics

Typical SEM images of the as-fabricated Inconel 718 sample are shown in Figure 7. Figure 7a clearly displays that the arc-like melt pools array in a layer-by-layer manner along the building direction (the X-Z plane and the coarser columnar dendrites grow across multi-layers parallel to the building direction because the bottom heat flow is almost uniformly transferred in the direction parallel to the negative *Z-axis* and perpendicular to the substrate. Figure 7a,b shows the clearly columnar dendrites in the melt pool’s fusion zone in the X-Z plane and fine equiaxial grains in the X-Y plane, respectively, which are consistent with previous studies [21]. 

When the high-energy laser beams focused on the pre-spread powder layer, an initial planar solid/liquid interface (Figure 8a) was formed with a temperature gradient perpendicular to the substrate After that, the first bulge appeared in front of the interface (Figure 8b) and tended to grow along the deposition direction, which resulted in the solute’s lateral discharge and accumulation at the root of the bulge. Solute enrichment reduced the solidification equilibrium temperature and made other bulges easier to form (Figure 8c) and continue to grow in the opposite direction of heat flow, eventually forming columnar dendrites. Aggregation of refractory solute elements discharged from the solidified liquid in interdendritic regions facilitated the nucleation of the Laves phase. The element distribution maps (Figure 9) confirm the presence of the aggregation of refractory elements in the interdendritic region again.Figure 9a is the STEM images of the SLM as-built sample in the X-Y plane, and Figure 9b–f are the EDS maps of the distribution of different elements. It can be seen from Figure 9 that the γ-matrix constituent elements Ni (Figure 9f), Fe (Figure 9e), Cr (Figure 9d) are uniformly distributed in the γ-matrix. It is worth noting that elements with large atomic radii such as Nb (Figure 9b) and Mo (Figure 9c) gather in the interdendritic region (the bright white region in Figure 9a). The segregation of these Nb and Mo elements leads to the formation of fragile Laves phase, which further confirms the accuracy of previous phase composition analysis and element distribution test results.

Zhang et al. [33] conducted a numerical simulation of the heat flux field of laser micro-molten pool and explored the influence of heat flow field on SLM technology. The results show that the surface tension caused by the temperature gradient on the surface of the molten pool drives the Marangoni convection, making the flow state of the SLM-process fluid mainly outward convection. Although the heat flow of a single molten pool varies from the edge to the center, in general, the main direction of the temperature gradient is almost consistent with the Z-axis, so columnar dendrites can grow through multi-layer structures in the Y-Z plane.

Many studies show that the microstructure of the SLM- specimens at different E all have columnar dendrites structure growing almost parallel to the Z-axis, showing obvious epitaxial growth characteristics. The characteristics of columnar dendrite on the X-Z plane and equiaxed crystals on X-Y plane lead to the obvious anisotropy of the as-built sample. In general, the coarser the columnar dendrites in the X-Z plane, the larger the diameter of equiaxed crystals on X-Y plane are. Therefore, SEM images of equiaxial crystals in X-Y plane were selected in the present study to elaborate on the effect of the input E on microstructures of SLM-Inconel 718 specimens, as shown in Figure 10. Obviously, with the increase of E, the morphology of equiaxed crystals is almost unchanged, but the grains’ diameter gradually decreases, grains become more refined. In order to quantify the relationship between the E and the grain’s diameter, the distance between adjacent grains is measured ten times, and the average value is taken as the grain size under the corresponding E, representing the diameters of the columnar dendrites in the X-Z plane. The results are shown in Table 5. 

As the input E increased, the epitaxial growth characteristics of the columnar dendrites in the shaped sample were gradually enhanced. Since the grain growth process mainly relied on grain boundary migration, the temperature and solidification rate had an important influence on grain boundary migration. Because of the anisotropy of the material, the grains can grow preferentially at relatively higher temperature gradients and higher solidification rates, and the preferred direction of grain growth was different due to the different heat flux density. Generally, most of the heat generated by the laser during SLM was dissipated through the substrate and the previously solidified material, which also provided thermodynamic conditions for the formation and growth of columnar dendrites in the opposite direction of heat dissipation. In addition, the nucleation and growth patterns of columnar dendrites were closely related to the homologous kinetics theory of the liquid in the molten pool during solidification. As the scanning speed decreased, the heat dissipation rate in the molten pool slowed down due to a longer laser irradiation residence time at a local region. The relatively prolonged cooling time provided strong kinetic conditions for the nucleation and subsequent growth of columnar dendrites. According to previous studies [34], a higher G/R value is the driving force for columnar dendrites to nucleate and grow along the crystal plane and direction that most facilitates crystal growth (where G is the temperature gradient in the molten pool, R is the growth rate of crystals in the molten pool). As the increasing of input E, the pool absorbed more energy, the temperature of the liquid phase in the pool raised and the pool solidification time was longer. That is to say, with the increase of temperature gradient and the decrease of solidification rate in the micro-pool, the G/R value increases. The higher G/R value provides strong thermodynamic conditions for the orientated continuous growth of columnar dendrites and promotes its uniformity and refinement. 

SLM forming of Inconel 718 samples were carried out based on a complete melting/solidification. The heterogeneous nucleation of γ crystal and the following continuous growth of columnar dendrites contribute to the formation of columnar dendrite structure. Based on the Gibbs-Thomson law, the dendritic tip temperature $T_{t}$ can be determined by [35]:(6)$$T_{t}=T_{M}+eC_{Li}^{*}-\frac{RT_{M}^{2}}{\Delta H_{f}}\cdot \frac{V_{t}}{V_{0}}$$
where *T_M_* is the fusion point of the matric, *e* the slope of liquidus line $C_{L_{i}}^{*}$, the solubility of the liquid at the solid-liquid interface, $\Delta H_{f}$ the material potential heat, $V_{t}$ the dendritic tip growth rate, and
$V_{0}$ is a dynamical constant. From the literature [36], *V*_t_
is dominated by the laser scanning speed according to the following formula:(7)$$V_{t}=v\cos\theta$$
where θ is the angle between the vectors $V_{t}$ and *v*. It can be seen that *V_t_* and *v* are in a proportional relationship. Thus, the higher scanning speed *v*, the faster the dendrite growth rate is, which proves the fact that columnar dendrites formed at a higher scan speed of 1000 mm/s are obviously coarser than that of 600 mm/s. Furthermore, SLM forming process is based on high energy laser beam repeated rapid melting/solidification process. When the scanning speed is too high, a large amount of residual stress is accumulated in the formed sample, which finally induces the defects between the columnar dendrites.

Figure 11 shows EDS analysis results for the contents of various elements in the center of the columnar dendrite microstructure under different E. It is found that the content of Ni element in the center of the columnar crystal rapidly increased from 49.56% to 54.52% as E increasing from 103 J/mm^3^ to 190 J/mm^3^. According to the solid solution theory [37], other refractory alloy elements of the γ matrix were replaced by a large number of Ni elements. It is known from XRD analysis that the constituent elements of the solidified matrix phase are Cr, Fe, Ni. Among them, the atomic radius of Ni element is the smallest, and the solid solution of Ni atom into the matrix phase causes negative lattice distortion, which reduces interplanar spacing of the matrix phase, further confirming the theoretical inference in Section 3.2.

### 3.4. Microhardness

The measured microhardness and its distribution on the polished cross-section of the SLM-fabricated parts are shown in Figure 12. All hardness values in Figure 12 are the average of hardness values of the same sample tested ten times under the same conditions. As the increasing of input E from 103 J/mm^3^ to 190 J/mm^3^, the mean microhardness increased significantly from 290.2 HV_0.2_ to 348 HV_0.2_ and the fluctuation of hardness distribution decreased synchronously. In general, the hardness value distribution of the formed sample was positively correlated with the input E. As the applied E increased, the average microhardness value of the sample increased gradually, and the fluctuation was gradually stable.

The reason for the rapid increase in microhardness is the denser internal structure and finer columnar dendrites caused by the increasing of input E. In addition, during the SLM forming process, as the heat was accumulated, a solid phase transition occurred to induce precipitation of a new reinforcement phase γ″. Since the structures of the reinforcement phase γ″ and the matrix phase γ were different, the reinforcement phase γ″ hindered the dislocations motion of the matrix during the deformation processing, producing a cutting action and a wraparound effect (dislocation loop), which resulted in greater strength and hardness of the parts. Furthermore, the solution of Ni atoms in the matrix also caused negative lattice distortion, which impeded the push of dislocation during the deformation process of the sample, and ultimately improved the strength and hardness of the sample [37]. In general, the microhardness of SLM-fabricated parts was enhanced by the comprehensive effect of grain refinement, densification, precipitation strengthening of reinforced phase″, and solid solution strengthening with the increase of applied E. 

### 3.5. Wear Performance

The variations of the coefficient of friction (COF) and wear rates of the SLM as-built samples at different E are shown in Figure 13 and Figure 14, respectively. Obviously, the input E played an important role in the sample’s wear properties. With the increment of E from 103 J/mm^3^ to 190 J/mm^3^, the mean COF reduced from 0.40 to 0.29 and the corresponding wear rate from 10.31 × 10^−4^ mm^3^/Nm to 5.67 × 10^−4^ mm^3^/Nm. In addition, the COF value of the largest 0.40 shows an apparent fluctuation phenomenon with the sliding time increasing under the lowest E of 103 J/mm^3^. At a higher E of 143 J/mm^3^, the mean COF value reduced to 0.33, resulting in a lower wear rate of 7.42 × 10^−4^ mm^3^/Nm. When the input E reached 190 J/mm^3^, the COF decreased gradually to a stable value of 0.28 with the lowest wear rate of 5.67 × 10^−4^ mm^3^/Nm. From the experimental results, the higher the input E, the smaller the COF value was, that is, the wear resistance of the material was improved.

During the wear test, the stainless steel grinding ball continuously slid on the surface of the SLM as-built Inconel 718 parts. In the case of lower E, because the sample had more holes on the surface and lower hardness value which cannot resist the intrusion of the grinding ball under the load pressure, a large amount of debris was peeled off under the mechanical shear stress of the rotation test, making the wear rate higher. At the same time, the microstructure was coarse and unevenly distributed, so the COF value was large and the fluctuation was severe. As E increased, the density and microhardness of the SLM-sample were greatly improved. The surface can effectively resist the intrusion of the grinding ball under the load pressure, the mechanical shearing effect was weakened. During the wear test, a protective adhesive tribolayer was formed to lubricate the friction pair, so the COF and wear rate of the sample were both reduced [38]. Uniform distribution of microstructure resulted in uniform distribution of hardness values, therefore, the COF value tended to be reduced and stable. Thus, with the increase of E, rolling friction replaced sliding friction as the main mechanism of material removal in wear tests, resulting in self-lubricating capacity [38].

## 4. Conclusions

Selective laser melting fabrication of Inconel 718 superalloy has been performed in this study. The effect of input laser volume energy density (E, J/mm^3^) on densification, phase composition, microstructural architectures, homogeneity and mechanical properties of the SLM as-built specimens were analyzed in detail. The main conclusions are summarized as follows:
(1)The surface morphology and densification of the SLM as-built Inconel 718 specimens were controlled by input E. Within the parameters selected in this study, with the increase of input E, surface holes and balling effect were gradually weakened and disappeared, the surface morphology became smoother, the scanning tracks tended to be smooth and continuous, and the sample densification level was improved steadily. Density increases from 95.75% at the lowest E of 103 J/mm^3^ to 99.15% at the highest E of 190 J/mm^3^. When the E was properly set, the near-full dense specimens with good metallurgical bonding and no critical defect can be obtained.(2)The microstructure of the SLM as-built specimens showed obvious orientational distribution characteristics. It was found that the coarsen columnar dendrites grew across multi-layers along the building direction in the X-Z plane and the fine equiaxed crystals were observed in the X-Y plane, because the main temperature gradient direction was almost consistent with the Z-axis although the heat flux of a single pool varied from edge to center. With the increasing of E, the columnar dendrites became finer. The elements with large atomic radii such as Nb and Mo clustered in interdendritic region. The microsegregation of ehese Nb and Ti elements resulted in reduced γ″ -phase content and the formation of brittle Laves phase.(3)By increasing applied E, the microhardness and wear resistance of SLM-fabricated parts have been significantly improved, and their fluctuation range has been reduced simultaneously. As E was changed from 103 J/mm^3^ to 190 J/mm^3^, the average microhardness increased significantly from 290.2 HV_0.2_ to 348 HV_0.2_, average COF from 0.40 to 0.29 and the corresponding wear rate from 10.31 × 10^−4^ mm^3^/Nm to 5.67 × 10^−4^ mm^3^/Nm. The combined action of higher density, higher microhardness and friction protective layer improved the wear performance.


## Figures and Tables

**Figure 1 materials-14-00991-f001:**
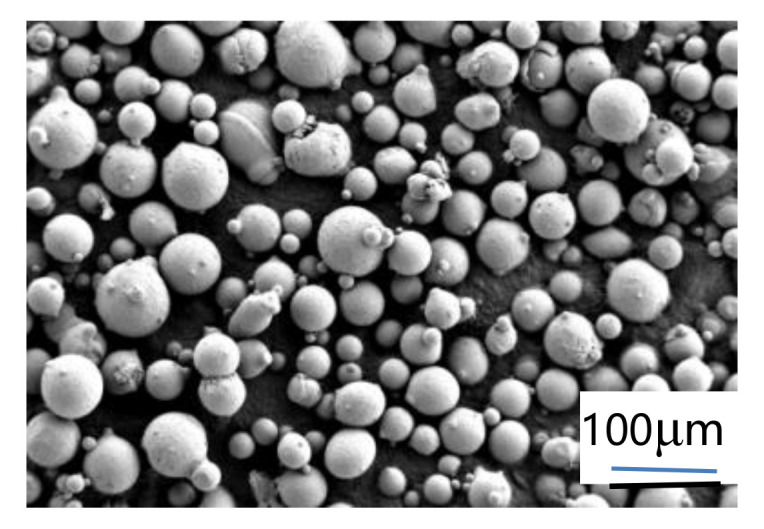
SEM image of Inconel 718 raw material powder.

**Figure 2 materials-14-00991-f002:**
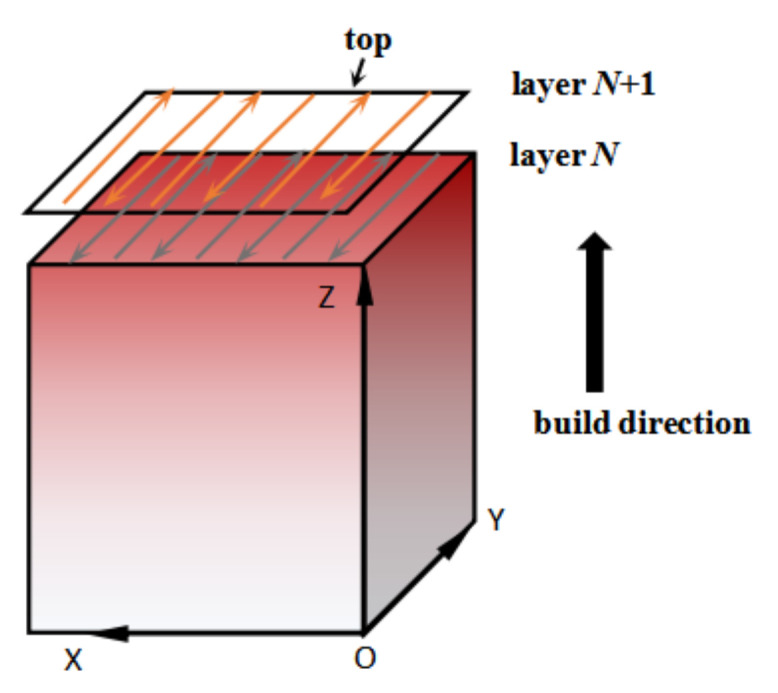
Laser selective melting scanning strategy schematic.

**Figure 3 materials-14-00991-f003:**
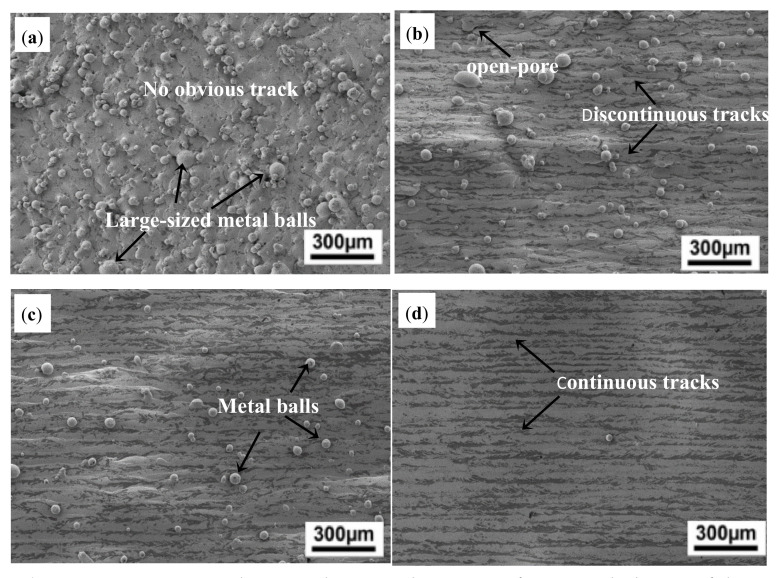
SEM images showing the typical upper surface morphologies of the as-built Inconel 718 specimens at various laser parameters: (**a**) E = 103 J/mm^3^; (**b**) E = 143 J/mm^3^; (**c**) E = 171 J/mm^3^; (**d**) E = 190 J/mm^3^.

**Figure 4 materials-14-00991-f004:**
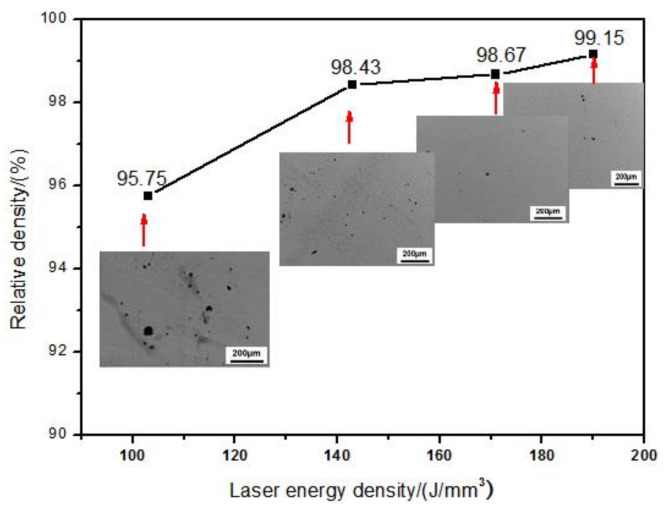
Effects of linear input E on the densification level of SLM Inconel 718 samples. OM images of the corresponding sample cross-sections are contained.

**Figure 5 materials-14-00991-f005:**
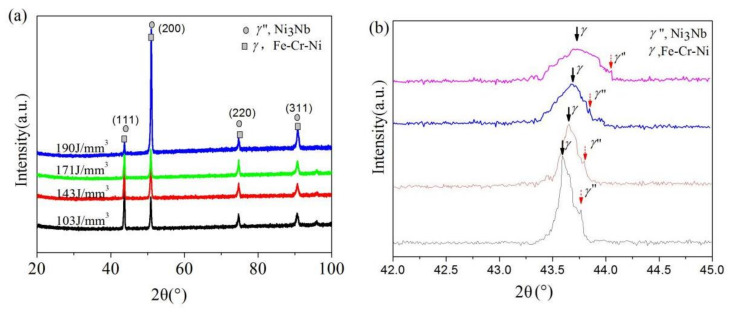
XRD patterns of SLM formed Inconel 718 parts under different laser energy densities obtained in a wide range of 2θ = 20°–100°(**a**) and obtained in a small range of 2θ = 42°−45°(**b**).

**Figure 6 materials-14-00991-f006:**
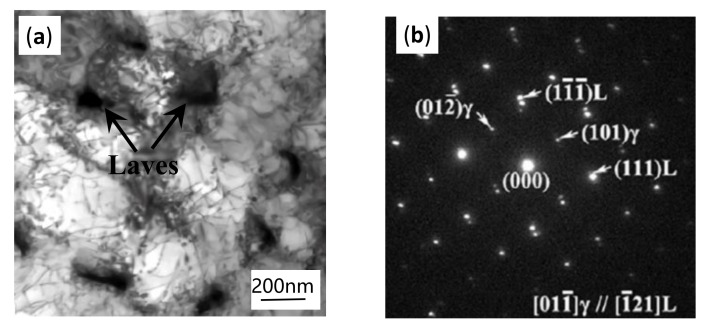
(**a**) TEM image displaying the irregular precipitates in the X-Y plane and (**b**) SAED patterns of Laves and γ

**Figure 7 materials-14-00991-f007:**
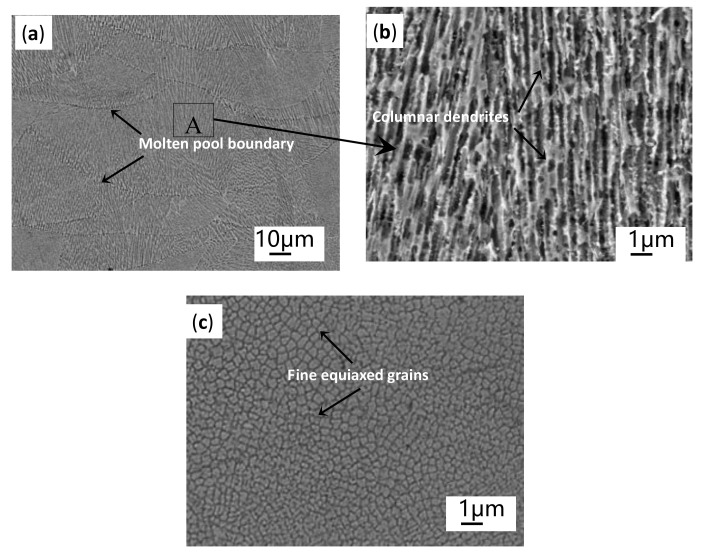
Typical SEM micrographs of as-built sample:(**a**) dendrites growing through multilayers on cross-section (X–Z plane); (**b**) clear columnar dendrites on the vertical section in the melt pool’s central fusion zone, which is the high-magnified micrograph of the area A in Figure 6a; (**c**) the fine equiaxed grains on the X-Y plane in the melt pool’s central fusion zone.

**Figure 8 materials-14-00991-f008:**
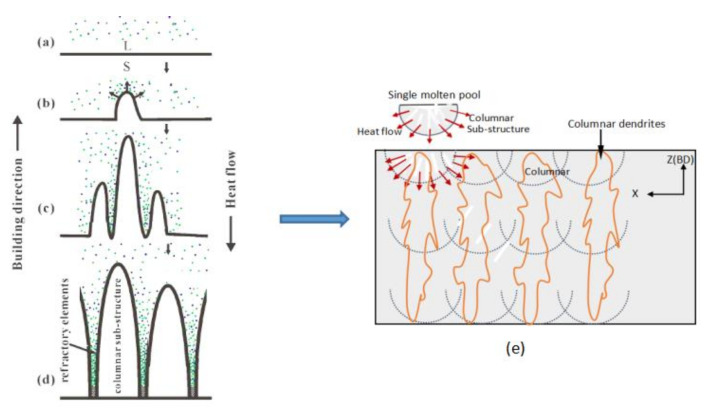
Schematic diagram of the growth process of columnar dendrites:(**a**) initial plane solid/liquid interface;(**b**) forming of first bulge on the interface;(**c**) bulges grow along building direction;(**d**) final columnar dendrites; (**e**) microstructures in the X-Z plane.

**Figure 9 materials-14-00991-f009:**
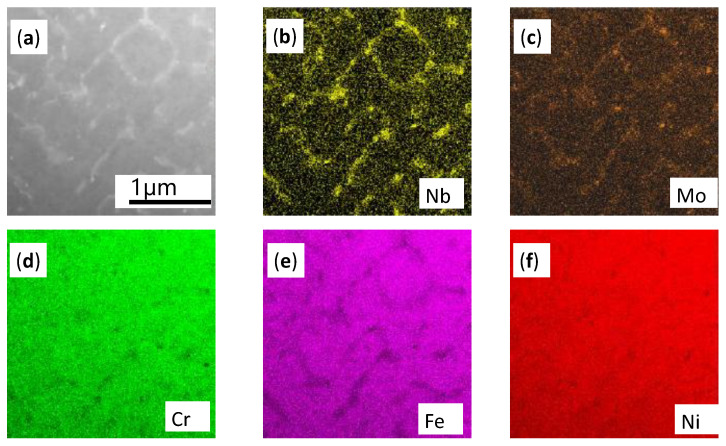
STEM images of the SLM as-built sample in the X-Y plane for compositional analysis. (**a**) Local TEM Morphology; (**b**–**f**) Distribution of Nb, Mo, Cr, Fe and Ni corresponding to Figure 9a, respectively.

**Figure 10 materials-14-00991-f010:**
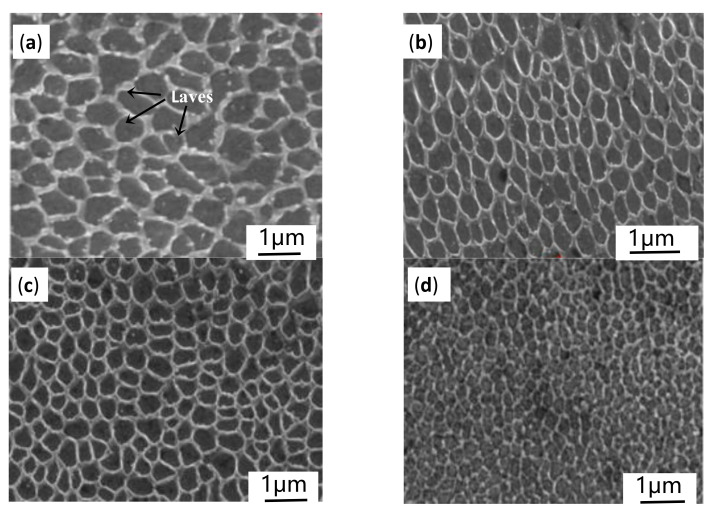
SEM images on the X-Y plane in the melt pool’s central fusion zone under different laser energy density, laser powers and scan speeds: (**a**) 103 J/mm^3^, 90 W, 1000 mm/s; (**b**) 143 J/mm^3^, 100 W, 800 mm/s; (**c**) 171 J/mm^3^, 90 W, 600 mm/s; (**d**) 190 J/mm^3^, 100 W, 600 mm/s.

**Figure 11 materials-14-00991-f011:**
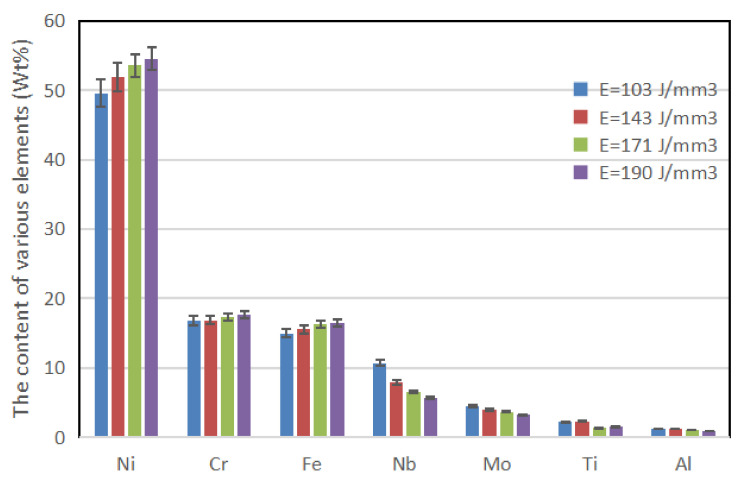
EDS analysis showing chemical compositions at the core of columnar dendrites under different E.

**Figure 12 materials-14-00991-f012:**
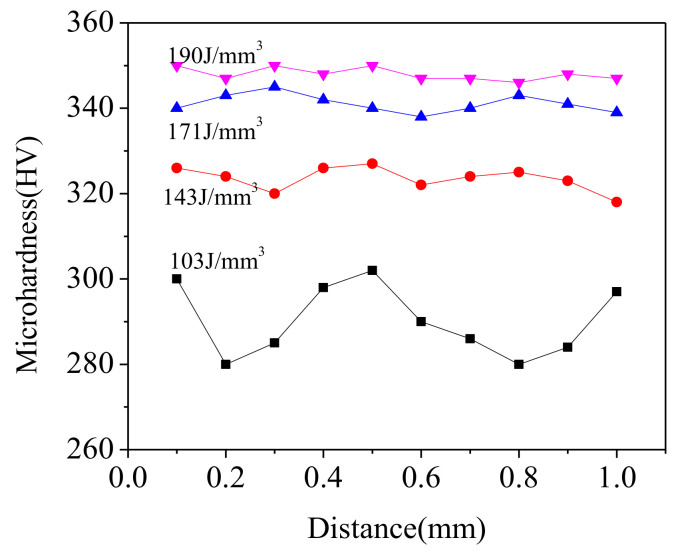
Hardness distribution of SLM as-built samples at different E.

**Figure 13 materials-14-00991-f013:**
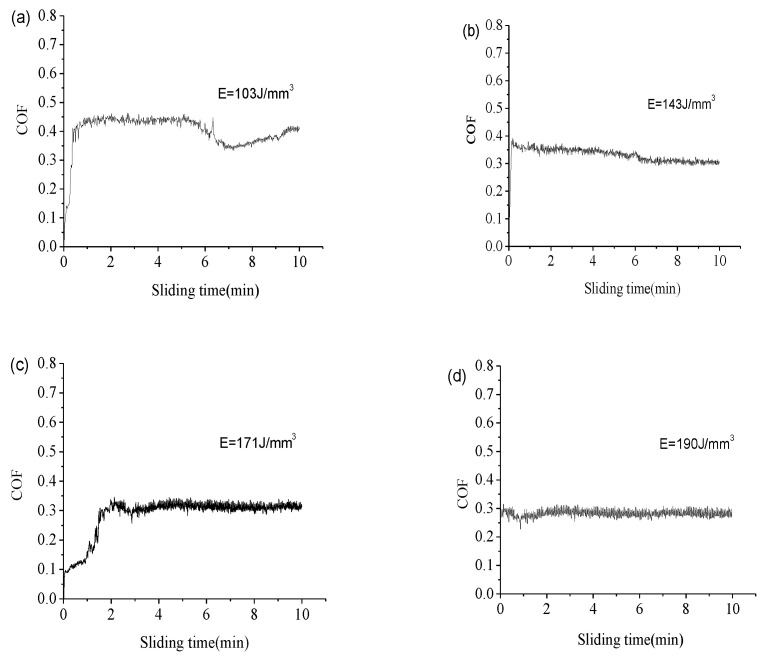
COF of the SLM as-built parts under different E over the sliding time. (**a**) COF of the SLM as-built parts under E = 103 J/mm^3^ over the sliding time; (**b**) COF of the SLM as-built parts under E = 143 J/mm^3^ over the sliding time; (**c**) COF of the SLM as-built parts under E = 171 J/mm^3^ over the sliding time; (**d**) COF of the SLM as-built parts under E = 190 J/mm^3^ over the sliding time.

**Figure 14 materials-14-00991-f014:**
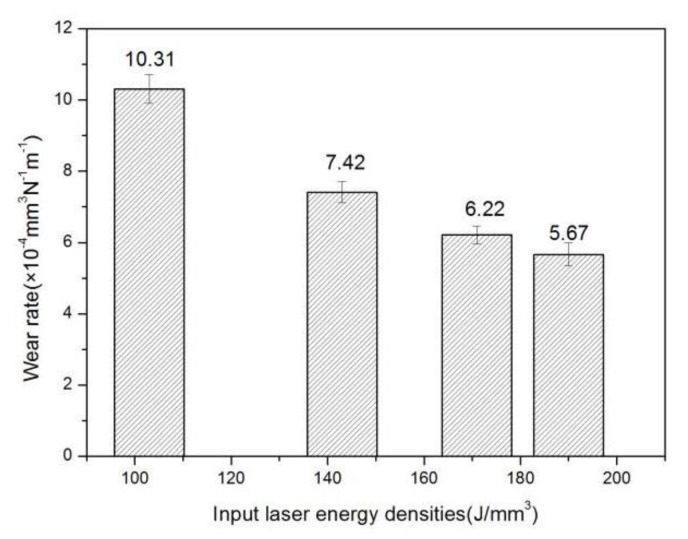
Wear rates of the SLM as-built parts under different E.

**Table 1 materials-14-00991-t001:** Chemical compositions of raw Inconel 718 powders (wt%).

Ni	Cr	Fe	Nb	Mo	Al	Ti	C
Balance	18.10	14.98	10.96	4.62	1.58	2.80	0.08

**Table 2 materials-14-00991-t002:** Processing parameters set in SLM processing.

Parameter	Value
Laser spot diameter /(D, μm)	50
Powder layer thickness/(t, μm)	25
Hatch distance/(h, μm)	35
Laser power/(P, W)	90, 100, 90, 100
Laser scanning speed/(v, mm/s)	1000, 800, 600, 600
Laser energy density/(J/mm^3^)	103, 143, 171, 190

**Table 3 materials-14-00991-t003:** XRD data of identified peaks for Inconel 718 parts.

Sample (J/mm^3^)	*γ*		*γ*″	
	2θ (°)	Intensity	2θ (°)	Intensity
103	43.591	1698	43.765	1056
143	43.653	1676	43.796	1035
171	43.673	1149	43.847	959
190	43.743	1063	43.938	950

**Table 4 materials-14-00991-t004:** Lattice constant values at different E based on Rietveld refinement.

Input Laser Energy Density (E, J/mm^3^)	Γ (fcc)		*γ*″ (bct)		R-Factors
	Cell Parameter (Å) a = b = c	Weight%	Cell Parameter (Å) a = b ≠ c	Weight%	
103	3.6039 ± 0.0014	92.54	a = 3.6259 ± 0.0020 c = 7.4013 ± 0.0016	7.46	R_p_ = 5.96 R_wp_ = 7.54 R_exp_ = 6.38 GOF = 1.40
143	3.5993 ± 0.0014	92.57	a = 3.6237 ± 0.0016 c = 7.4000 ± 0.0012	7.43	R_p_ = 6.10 R_wp_ = 7.75 R_exp_ = 6.49 GOF = 1.45
171	3.5988 ± 0.0001	92.74	a = 3.6214 ± 0.0006 c = 7.3983 ± 0.0006	7.26	R_p_ = 5.49 R_wp_ = 7.51 R_exp_ = 6.23 GOF = 1.39
190	3.5972 ± 0.0001	92.82	a = 3.6206 ± 0.0001 c = 7.3868 ± 0.0007	7.18	R_p_ = 7.22 R_wp_ = 9.75 R_exp_ = 7.19 GOF = 1.84

**Table 5 materials-14-00991-t005:** The relationship between E and the diameters of the columnar dendrites in the X-Z plane.

**Input laser energy density (E, J/mm^3^)**	103	143	171	190
**Diameter of columnar dendrites in the X-Z plane(μm)**	0.53	0.42	0.31	0.27

## Data Availability

The data presented in this study are available on request from the corresponding author.

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
