# Peer review of "Formability, Microstructure and Properties of Inconel 718 Superalloy Fabricated by Selective Laser Melting Additive Manufacture Technology"

_materials, 2021, doi:10.3390/ma14040991_

Round 1

Reviewer 1 Report

It is a correct scientific and technological work about the microstructure and properties of Inconel718 Superalloy. The alloy has been fabricated by laser melting additive manufacture technology.

I recommend the publication of this manuscript in the indexed journal Materials after major revision.

Comments:

XRD analysis should be improved. The authors apply the Bragg equation. There are so many methods to obtain crystallographic parameters. Some of them are based on the Rietveld refinement. An alternative are linear methods as Williamson-Hall to determine the lattice parameters and the microstrain.

How many EDX microanalysis have been performed? The results of the EDX microanalysis are given without statistical error.

Likewise, the authors should check the homogeneity of the superalloy Inconel718.

In the SEM micrographs is necessary the addition of scale bars

Figure 2 is not necessary. The related information should be given in the manuscript.

The authors should check typographic errors.

Reviewer 2 Report

Dear Authors, 

I carefully read your manuscript and I seriously thought about rejecting your work. But, finally, in my opinion, you have to get a chance for a significant improvement in your manuscript.

  1. Overall comment.

In general, your paper contains a lot of serious punctuation mistakes, misspellings, grammatical errors which make your manuscript hard to read i.e:

  • line 34: (Nevertheless ,its) - punctuation,
  • line 326: (in the present paper) - grammar/language (you mean presented?).
  • line 138: (the laser scanning tracks were clearly and tidy) - grammar

Another serious issue is connected with technical language errors: 

  • powder paving thickness instead of proper: layer thickness
  • the sample was inlaid instead of proper: mounted
  • two, the above mentioned are just examples - there is a lot of that kind of mistake. 

In my opinion, authors have to consider some language edition services or ask native speakers with experience in the technician language for help because, in that form, the paper is not acceptable. 

The next thing which makes this manuscript hard to read is a way of process parameters presentation:

  • line 85- "Based on a series of preliminary 85 experiments,the spot diameter was settled at 50μm, the scanning pitch h of 35μm, and the powder 86 paving thickness of 25μm;the laser power(P) was preset at 90, 100, 90,100W and scanning speed was 87 set at 1000,800,600,600 mm/s." - It has to be put in the table to make it easy to analyze. 
  • Figure 8 - charts are not described so it is impossible to guess what kind of sample was tested.
  • line 89: you put equation without number and explanation of used symbols - what is v ???  

Another issue is connected with physical values, in my opinion, authors have to use values that are more common and mostly used in additive manufacturing: (i.e. energy density in J/mm). 

        2. Abstract

The abstract is not very well written. It has no information about the novelty and background of this job. Also, the word "optimal" appears in line 20. Please put information about your optimization criteria.  
You described η as laser energy and then named it as energy density. 

        3. Introduction 

This part has to be significantly improved. There is no information about the available research results in the literature. You have to analyze other works and mention what other authors achieved (i.e. XY et al. [...] obtained the following results, where XZ et al. registered a slightly different phenomenon, etc.). In general, you have to more specific during the literature review which should lead to the research gap which you wanted to fill. 

       4. Materials and Methods

a) line 97: are you sure your method is high precision? Is it more precise than typical porosity analysis (using a microscopical analysis) or tomography? 

b) lines 116 and 119 pleas put units to these equations. 

      5. Results and discussion

a) I have serious doubt about your interpretation of the balling effect. You used a laser source with quite low power (100W) are you sure that your phenomenon is not connected with non-melted grains??? Please elaborate with that and add proper literature to prove your state. 

b) Based on your figure 4 it could be stated that further increasing linear laser energy density seems to be positive. Please elaborate with also that issue. 

c) In line 209 you sentenced: "the solidified structure still appeared as columnar dendrite, 209 but it became significantly longer and refiner." What was the base of that sentence??? 

d) Line 229: "As the input η is further increased, more heat generated by the laser irradiation will be absorbed by the molten pool, and the temperature of the liquid phase in the molten pool rises rapidly.  The increase of internal energy and thermodynamic potential brought about by the accumulation of heat provides the driving force for nucleation and growth of the epitaxial growth of columnar dendrites. Furthermore, a tension gradient on the liquid phase surface of the molten pool ,which was caused by the temperature gradient and chemical concentration gradient in the micro-melting pool , produced Marangoni convection and liquid-phase capillary force, promoting re-alignment of the solidified crystal nucleus which made the crystal structure distribution more uniform and finer."

Did you make some thermophysical analysis? Please cover your statement. 

d) line 292 - please put information about the mentioned "optimization of process parameter".

       6.Conclusions

After all corrections in your manuscript, which I pointer earlier please update your conclusions. 

       7. Final summary

As you can see, you have to make a lot of work to make your paper ready for publication. I believe my comments will be useful to improve your work. Please take your time to make proper quality (high quality - of course) corrections to all mentioned issues. 

Round 2

Reviewer 1 Report

It is a correct work about the formability, microstructure and properties of Inconel718 Superalloy fabricated by SLM (selective

laser melting) additive manufacture technology. A major revision is necessary.

Comments:

XRD: The authors apply the Rietveld refinement. The authors should give the goodness of fit (GOF) parameter. Likewise, correct Rietveld analysis should provide the % of each phase (there are some XRD diffraction patterns with two coexisting phases).

In table 4 the authors should remark the crystallographic phase of the given lattice parameters. Furthermore, the values and the errors should have the same precision (and the error 1 or maximum two significant numbers. As an example, 3.60391+-0.001427 should be 3.6039+-0.0014.

Information as “wide range of 2θ=20°ï½ž100°at the quick scanning speed of 4°/min” should be given in the Materials and Methods section, not in the experimental section.

How many hardness experiments were performed for each sample? A statistical standard deviation (or equivalent parameter) should be provided.

How many EDS compositional analysis were performed for each sample? A statistical standard deviation (or equivalent parameter) should be provided. Furthermore, I suggest EDS better than EDX because is a spectrometry based technique.

I suggest English revision by an expert.

There are a lot of typographic mistakes. The authors should accept all changes in the manuscript and to check all the text.

The authors should revise the reference format taking into account the guidelines of the journal.

Reviewer 2 Report

Dear Author, 

Indeed you made a significant improvement in your work. Now, in my opinion, it is ready for publication. 

I wish you all the best in your future scientific work. 

Author Response

Many thanks for reviewer's positive suggestions.